# The Relation between Consumer Perception and Objective Understanding of Front-of-Package Nutrition Labels (FOPNLs); Results from an Online Representative Survey

**DOI:** 10.3390/nu16111751

**Published:** 2024-06-03

**Authors:** Emmanuella Magriplis, Georgios Marakis, Demosthenes B. Panagiotakos, Aspasia Samona, Sotiria Kotopoulou, Dimitris Kouretas, Theodoros Smiliotopoulos, Michail Chourdakis, Antonis Zampelas

**Affiliations:** 1Laboratory of Dietetics & Quality of Life, Department of Food Science and Human Nutrition, Agricultural University of Athens, Iera Odos 75, 11855 Athens, Greece; skotopoulou@efet.gr (S.K.); teosmiliotopoulos@gmail.com (T.S.); azampelas@aua.gr (A.Z.); 2Hellenic Food Authority, Leoforos Kifissias 124 & Iatridou 2, 11526 Athens, Greece; gmarakis@efet.gr (G.M.); asamona@efet.gr (A.S.); 3Department of Nutrition & Dietetics, School of Health Sciences & Education, Harokopio University, 17776 Athens, Greece; dpanag@hua.gr; 4Department of Biochemistry-Biotechnology, University of Thessaly, 41334 Larisa, Greece; dkouret@uth.gr; 5Laboratory of Hygiene, Social & Preventive Medicine and Medical Statistics, School of Medicine, Faculty of Health Sciences, Aristotle University of Thessaloniki, 54124 Thessaloniki, Greece; mhourd@gapps.auth.gr

**Keywords:** FOPNL, NutriScore, NutrInform Battery, FOPNL preference, FOPNL objective understanding

## Abstract

Background: This study investigates the efficacy of Front-of-Pack Nutrition Labels (FOPNLs) as a cost-effective tool for improving dietary choices among Greek consumers. The purpose of the study was to investigate Greek customers’ preferences and comprehension of commonly used European FOPNL schemes. Methods: The Hellenic Food Authority and the Agricultural University of Athens performed a representative online survey in March 2022, titled “The Role of Nutritional Labelling in Public Perception and Food Procurement.” Consumers responded to a questionnaire separated into two parts. Part one included (i) personal, sociodemographic information, and (ii) subjective opinions on the FOPNL schemes, and part two comprised (iii) an objective understanding of NutriScore and NutrInform Battery, using 15 different foods. Participants were randomly allocated to these groups, and general mixed models were used for analysis. Results: A total of 1389 adults completed the first part of the survey, and 74.8% completed the second part. The Multiple Traffic Lights scheme was the preferred FOPNL, chosen by 48.4% of respondents, compared to 19.7% for NutrInform Battery and 12.3% for NutriScore. However, the mean objective assessment score was highest for NutriScore (5.8 ± 2.3) compared to NutrInform Battery (5.4 ± 1.9). Conclusion: The results highlight the necessity for comprehensive nutrition education programs by showing a considerable gap between subjective preferences and an objective understanding of nutrition labels.

## 1. Introduction

Nutrition labelling has been recommended by the World Health Organization (WHO) [1,2] as a cost-effective means to empower individuals in making healthier food choices at the point of purchase and hence improve dietary intake at the population level [3,4]. In the European Union (EU), a nutrition declaration table has been compulsory since 2014 [5], yet consumers infrequently consult it due to its complexity [6,7,8], since it requires a high level of health and nutrition literacy and time to interpret the information [9,10]. A recent review by Chrissini and Panagiotakos (2021) indicated that a large proportion of the studied populations in different countries possess inadequate health and nutrition literacy, which in turn is associated with higher intake of Western diet-related foods, unhealthy weight, and adverse health outcomes [11]. Front-of-Pack Nutritional Labelling (FOPNL), as a form of supplementary nutrition information, has emerged as a means to provide more visible and simplified nutritional information on the front of packaged foods, to facilitate consumers—particularly those of lower health and nutrition literacy—in the interpretation of the nutrient declaration and make conscious, healthier, and sustainable food choices faster [10,12]. It is noteworthy though that FOPNL can also help consumers with adequate health literacy to discriminate products with varying nutritional composition [8].

Various FOPNL schemes have been developed [10,13]. They can be broadly categorized as reductive (i.e., non-interpretative) or evaluative (i.e., interpretative). Reductive schemes are “nutrient-specific” and repeat some of the numerical information from the mandatory nutrition declaration in a neutral, non-evaluative way (e.g., Reference Intake scheme (RI), NutrInform Battery). The evaluative schemes are based on nutrient profile models. In Europe, such systems can be (a) color-coded “nutrient specific” (e.g., Multiple Traffic Lights (MTL), Evolved Nutrition Label (ENL)), or (b) “summary indicator”, presented either as a graded rating or score of the overall nutrient profile of a product (e.g., NutriScore) or as a positive “endorsement” logo (e.g., Keyhole symbol, Choices logo) that is applied only to those foods complying with certain nutritional criteria. The functional and visual aspects that form the fundaments of different FOPNL systems have been previously described [14] using the Funnel model as a tool.

Multiple FOPNL schemes coexist in the EU and even within the same country [15], as is the case in Greece, where RI, and to a lesser extent ENL and Nutriscore, can be found on the packages of different foods. Within the framework of the EU “Farm-to-Fork strategy” [16], the European Commission intends to propose a harmonized mandatory FOPNL scheme, to minimize the risk of coexistence of multiple FOPNL schemes that can create potential confusion to consumers [17,18] and obstacles to the free movement of goods in the EU.

Systematic reviews have examined the effect of FOPNL on consumer behaviors [12,19] and industry response [12,19]. One of the early systematic reviews and meta-analyses of randomized studies on FOPNL reported an estimated 18% increase in the proportion of consumers choosing a healthier food product in the presence of an FOPNL scheme [19]. A later systematic review, including 60 studies, showed that labelling decreased total energy and fat intake, increased vegetable intake, and led to an 8.9% decreased sodium content in manufactured products [12]. Since the publication of this review, a growing number of studies and reviews have consistently confirmed this in adults [12,13,15,20]. Even though numerous studies have compared the relative performance and impact of different types of FOPNL schemes, mixed results have been obtained, depending on the types of FOPNL schemes tested, on the methodology used (e.g., objective vs. subjective consumers’ understanding), and on various socioeconomic and cultural aspects [10,18,21,22,23]. Subsequently, there is still no consensus as to which FOPNL scheme is most effective in influencing consumers’ choices, irrespective of their preference and subjective understanding [10,24].

Despite this rapid evolution and innovation in the field of FOPNL in Europe, Greece has not developed or officially yet endorsed a specific FOPNL scheme. It is known that consumer interpretation and utilization of FOPNL may be dependent on sociocultural context and differs across cultures or countries [25]. Two studies have evaluated different FOPNL schemes in adult Greek populations, with contradictory conclusions. The study by Kontopoulou and colleagues in 2021 evaluated the objective consumer understanding of different FOPNL schemes and provided evidence that Nutriscore, compared to RI, helps Greek consumers identify the nutritional value of the products [26]. On the other hand, the cross-country study that included a Greek cohort [23] evaluated the subjective consumer understanding and liking of different FOPNL schemes and showed that NutrInform Battery, which is a variant of RI, is the preferred FOPNL scheme in helping consumers understand information in a relevant way. Because of the conflicting conclusions of the studies, it seems imperative that more relevant studies on these two FOPNL schemes, Nutriscore and NutrInform Battery, are needed in Greece.

The aim of this online survey was primarily to explore consumers’ preference, knowledge, and purchase intention of products with regard to frequently used FOPNL schemes in Greece. The secondary aim was to assess consumers’ objective understanding of the two FOPNL schemes Nutriscore and NutrInform Battery in relation to the Nutrition Declaration Table, using an experiment with examples of commonly consumed foods in Greece from five main food groups (dairy/non-dairy alternatives, oils, fruit juice/fruit drinks, and grains). The outcome of this study has the potential to guide evidence-based labelling policy in Greece and can be utilized in the ongoing debate at EU level with respect to the most informative and effective FOPNL system for European consumers.

## 2. Methods

This study was a population-based online consumer survey, entitled “The Role of Nutritional Labeling in Public Perception and Food Procurement” and conducted in March 2022. The Qualtrics Survey Platform (Qualtrics XM Platform TM) was used to upload the consumer survey and collect the data. All data were protected based on the Data Protection Policy of the Agricultural University of Athens, and participants were provided with the Data Protection Officer’s and the Principal Investigator’s emails for any queries and/or concerns regarding the study. A description of the study’s aims and objectives, underlining the confidentiality and the non-obligatory participation, was disclosed to all participants for their consent prior to survey commencement. Participants were invited to complete an online survey hosted by the Agricultural University of Athens (AUA) in conjunction with the Hellenic Food Authority (EFET). The study was approved by the Ethics Committee of AUA and the Management Board of EFET, according to national law.

### 2.1. Study Sample

The study aimed to include a representative sample for three population groups by age, specifically 18–24.9, 25–59.9, and 60+ (3 groups), based on the data tabulates in the latest available Hellenic census (2011). As per WHO STEPS surveillance, a minimum of 384 individuals are required for a 95% level of confidence (*Z* = 1.96), a 5% margin of error, and a conservative approach of 50% estimated prevalence of various risk factors, since no other previous information is available.
$$n=Z\frac{P(1-P)}{e^{2}}$$
where *Z* = level of confidence; *P* = baseline level of the indicators; *e* = margin of error.

A total of 1152 adults were required to achieve representativeness for three groups (384 × 3 = 1152) and a 1.0 design effect. The sampling framework used was the latest population census available when the study was conducted. The population was grouped in the three age groups for which we aimed to achieve a representative sample, from which specific rates were calculated. These were applied to the final study sample and calculated in order to obtain the final number per age group required (n_1_ = 124 for 18–24.9, n_2_ = 642 for 25–59.9, and n_3_ = 289 for 60+). The design effect was chosen based on the combination of methods used to acquire the sample. Specifically, the researchers (i) sent invitations via email, using an undisclosed recipient system, to available predefined panels; (ii) posted the survey on various media, including Facebook and Instagram; (iii) publicized through the university and academic websites; and (iv) publicized through the website of a non-profit organization that focuses on adults aged 50+ years. In all cases, the promotion emphasized to individuals that were reached to spread the word. Data were frequently checked for total number and age distribution, to employ corrective actions and promotions through the web and word of mouth accordingly. Study participants included all adults 18+ who were able to read and had cognition. Exclusion criteria, other than age, were individuals that had incomplete data, responding to less than 80% of the questionnaire.

A total of 1934 individuals reacted to the emails and/or the posts on social media and clicked on the survey’s link, and 1389 responded to the questionnaire (72% response rate). Participants that had completed <80% of the questionnaire (part 1) and those <18 years of age were excluded (complete information of the consumer online survey enrollment can be seen in Figure 1). An adequate number of participants for national representativeness was achieved for generalization in total and for the younger age groups (18–24.9 and 25–60 years) but not for those 60+ years of age (n = 206 compared with the aim of 289).

### 2.2. Survey/Questionnaire Development

Upon consent, participants were asked to report personal data (date of birth, gender, area of residence), sociodemographic information (occupational status, educational level, marital status), household size (number of members residing in the same home and number of children specifically), smoking status (smoker, non-smoker, ex-smoker), anthropometric information (weight and height), and health indicators (presence of chronic disease(s): hypertension, Type 2 Diabetes Mellitus, high cholesterol, cardiovascular disease, cancer—any type). For the health indicators, participants were given the possibility of multiple selections. Consumers were also asked to report whether they were following a vegan or a vegetarian diet, whether they follow a special diet (energy restricted, low in salt, low in fat, and/or low in sugar), and if they had an educational background in nutrition, to avoid response bias. As per the study aims, the survey was then separated into two sections: the nutrition perception section (subjective) and the nutritional assessment (experimental, for objective evaluation) section. Construct validity was assessed by an expert team consisting of 2 nutritionists, one from academia and the second from the Hellenic Food Authority, a nutritional epidemiologist, an epidemiologist with expertise in methodology, and a consumer scientist. The survey was pilot tested using a convenience sample of 50 individuals before launching to ensure the understanding of all questions. Survey questions (5 in total) were reworded according to feedback to improve clarity. The reliability of the survey was not evaluated, since the questionnaire aimed to assess perception, where no specific measurable outcome is available. In terms of the understanding, subjective section errors were minimized through random allocation.

#### 2.2.1. Nutrition Perception Section (Subjective)

This section selected subjective information regarding consumer habits, self-estimated level of nutrition knowledge (including dietary recommendations for health) along with questions pertaining to awareness, and perception of frequently used FOPNL schemes on food packages. A detailed description of the FOPNL schemes used in the survey is presented in Table 1, with a brief explanation in each case. Participants were asked to report their level of agreement related to nutritional labeling overall (its importance, reading of the ingredient list and nutritional labels, search for nutritional claims on food packages, search for the products’ nutrient content, preference to select food with nutritional claims, comparison of prices between products, importance of price compared to nutritional value, and avoidance of foods perceived as unhealthy irrespective of label), and their subjective understanding of selecting the healthiest product based on the information provided. Responses were based on a five-level agreement scale provided for all questions, ranging from strongly disagree to strongly agree (somewhat disagree, neither agree nor disagree, agree). For the FOPNL schemes, participants were first provided with the description of various types of labels (reductive versus evaluative; nutritional information per portion or by 100 g of product) and reported their level of agreement. They were then asked (i) if they had noticed any of the displayed FOPNL schemes on foods they had bought to date (NutriScore, NutrInform Battery, Healthy Choice, none) and (ii) to select the FOPNL scheme (as previously stated as well as MTL) they would like to see on the front of the package (the FOPNLs included can be seen in Appendix A).

#### 2.2.2. Nutritional Assessment Section (Objective)

The second part included an experiment for the objective assessment of the consumers’ understanding of two FOPNL schemes as per the study’s aim, a summative colored (NutriScore) and a non-evaluative monochromatic (NutrInform Battery) scheme. The control used was the Nutrition Declaration Table, hence forming three groups in total. The Nutrition Declaration Table was set as the control, since most participants have been exposed to this for many years, and it is compulsory on the package. Fifteen (15) specific foods were chosen from various food groups. Participants that completed the first part of the questionnaire were randomly allocated by the online system to one of three groups using the computer software (Qualtrics Survey Platform). The randomization process was uniformly applied for the total population and all age groups. Statistical tests were performed to ensure that no significant allocation group differences were present (for age group, nutrition-related knowledge, and sex). The first group was provided with the Nutrition Declaration Table (Group A), the second group with Nutriscore (Group B), and the third with NutrInform Battery (Group C). All groups had the same foods, and these were delivered in the same order.

In Group A, the nutritional information was presented in the form of the Nutrition Declaration Table, as provided by the manufacturer (per 100 g or per 100 mL). For Group B and Group C, FOPNLs were calculated based on the nutritional information and ingredient list provided by the manufacturer. All calculations were performed by a trained dietitian using specific validated online databases for each FOPNL system per 100 g of food product to maintain consistency and for optimal direct between-group comparisons. Participants were exposed to the Nutrition Declaration Table or the FOPNL scheme for each food product, one by one, and they were asked to review the information and select the response they believed was most suitable in terms of consumption. Possible responses included (i) may consume ad libitum, (ii) must consume less frequently, (iii) must consume a smaller portion, (iv) must consume a smaller portion and less frequently, and (v) must avoid consumption. The questionnaire pilot tested by sending the link to a convenience sample (a total of 20 responses were received). Further corrections were made based on the feedback received. It was then uploaded as previously described.

#### 2.2.3. Foods Used in the Questionnaire

Five food categories were selected based on the following criteria: (i) usual consumption (oil and cheese), (ii) product availability on the market in terms of variety of products and plant-based alternatives for greater sustainability and health perception (yogurt), and (iii) variability in nutritional quality either related (puff pastry of the grain group) or unrelated to energy and fat content (e.g., feta cheese compared to reduced salt feta cheese, and whole wheat compared to refined tortillas of the grain group). Details on the foods used are found in Appendix A.

The nutrition information of each food was obtained from random real products on the Greek market and was displayed next to the relevant questions of the online questionnaire, either as s front-of-package nutrition scheme (for Groups B and C) or as the Nutrition Declaration Table (for Group A). The actual food product was described, but its picture was not displayed to avoid potential food industry probed response bias (in any direction). No other quality indicators, list of ingredients, or other information was provided. Correct responses were based on current national dietary guidelines for health [27] and packaging as sold. For example, in the case of yogurt, one portion is equal to 200 g as sold in the usual packaging. Beverage intake and the effects of fruit juice compared to fruit drinks are still debatable [28]. Refrigerated 100% fruit juice, although it is considered healthy, should be consumed at a smaller portion irrespective of packaging, since half a glass of juice (120 mL) rather than one glass is the recommended intake [27]. The response “consume less frequently” was also found acceptable and was regarded as correct. For fruit drinks with added sugar, the correct response was “avoid consumption”, whereas for fruit drinks without added sugar, “consume smaller portion and less frequently” was considered as the correct response [29]. Acceptable correct responses for each food are presented in Appendix A, and the label condition of the foods can be seen in Appendix A by food category.

### 2.3. Statistical Analysis

Hypotheses were specified prior to data collection, and the analytic plan was decided prior to data analysis. Data were checked for responsiveness and totality to decrease potential response bias in the results. For this task, binary dummy variables were created for each section of the survey, highlighting the number of questions responded to by each of the participants. Participants that responded to less than 80% of the questionnaire were excluded from the analysis (less than 46 of the subjects in the first section and less than 12 in the second section).

Data were then described by age group, with categorical variables as relative frequencies (n, %) and continuous data as mean (SD) or median (range), following normal distribution assessment. Between-group differences were assessed with the chi square test for categorical variables, and one-way ANOVA for normally distributed continuous data or Kruskal–Wallis for skewed distributions. FOPNL preference with regard to sociodemographic variables was also assessed using the chi square test to visualize potential consumer profile differences (Appendix A). Allocated FOPNL between-group differences for the objective assessment score were also tested to ensure correct randomized allocation (data provided in Appendix A).

Cubic splines, following adjusted general mixed models (GLM), were used to present the predicted probability of consumer behavior and FOPNL preference, by subjective perceived nutritional level awareness of dietary guidelines for healthy eating (based on a 1–10 Likert scale). The model was adjusted for age group, gender, educational level, occupational status, and nutrition-related degree. Objective assessment responses were reported by group, and tabulation tests with Tukey–Kramer post hoc analysis were conducted to present between-group differences for each food. A value of “1” was given for each correct (or acceptable) food response, and a total score was derived for each participant, with the possible score ranging from 0 to 15. The correct responses are depicted in Appendix A. The total number of participants responding acceptably was tabulated. The total objective assessment score for each group was derived in total and aggregated by age group, weight status, disease status (defined as individuals reporting being diagnosed with hypertension and/or Type 2 Diabetes Mellitus and/or hyperlipidemia and/or cardiovascular disease, and/or cancer) and background in nutrition. The latter was included to examine potential response bias due to nutrition knowledge. All analyses were conducted using the statistical package Stata 18 (StataCorp, College Station, TX, USA).

## 3. Results

In Table 2 and Appendix A, the main personal and sociodemographic characteristics of the participants are displayed. A high proportion of the respondents were 25–59.9 years of age, females, resided in the greater Metropolitan areas of Greece (Attiki and Thessaloniki), of higher educational level, employed full time, and single or married/cohabiting. Between-age-group differences were found for all variables other than female gender respondents. Furthermore, 74.8% of the respondents were also the main people responsible for the grocery shopping of the household.

The preferred scheme selected by most participants, irrespective of reported FOPNL awareness, was the MTL, where specific nutrients are displayed as per reference intakes (RIs) but with color coding (i.e., nutrient-specific evaluation scheme) (Figure 2). NutrInform Battery ranked second in total, and NutriScore third. The findings did not differ in the majority of the cases between different variables (sociodemographic, smoking, weight, disease status, or nutrition-related degree). However, consumers with an elementary educational level (≤6 years), despite the small sample, significantly responded that they preferred Healthy Choice (40% compared to 33.3% preferring MTL, 20% NutriScore, and 6.7% NutrInform; *p* < 0.001). Among those unaware of FOPNL schemes, second in ranking was Nutriscore and then Healthy Choice, while among those aware of FOPNL, NutrInform Battery ranked second, with a large difference from MTL (19.8% compared to 52.3%).

The proportion of individuals who observed the Nutrition Declaration Table and FOPNL schemes are displayed, in total and by level of FOPNL awareness, in Figure 3. The great majority of the population reported observing the Nutrition Declaration Table on the package in all cases (from 66.1% in unaware participants to 92.1% in those aware). As per the FOPNL scheme, the results indicated that most of the respondents were most exposed to Nutrinform Battery among all levels of awareness (Figure 3) compared to Nutriscore and Healthy Choice; from 19.3% and 17.9% lower in those aware to 26.8% and 30.3% in those unaware, respectively.

The predicted probability of the consumers’ behavior varied with every unit increase in perceived self-estimated nutrition knowledge (Figure 4). The predicted probability of buying a product due to a symbol or figure containing nutritional composition of food, decreased with each unit increase in perceived self-estimated nutrition knowledge, although those reporting good knowledge (>7) also avoided foods they perceived as unhealthy, irrespective of nutrition label. Participants with low self-perceived knowledge preferred energy intake to be presented by food portion. Most participants agreed on a nutrient-specific mandatory FOPNL scheme, but those with low nutrition knowledge (<4) also agreed on a mandatory FOPNL scheme that summarizes the nutritional value of the food. Lastly, the higher the nutrition knowledge, the greater the probability of the consumer being aware of the FOPNL schemes. It should be noted that in all cases, those rating 5–6 on the Likert Scale had no significant differences in their reporting in all variables examined.

As per the nutrition information understanding, all responses by allocated group can be seen in Appendix A. The mean objective assessment score was significantly higher among all participants allocated to Group B (NutriScore, *p* = 0.009), with a mean of 5.8 (2.3) compared to 5.4 (1.9) among all participants allocated to Group C (Table 3). No significant differences were found between the other groups. Although significantly higher mean objective assessment scores were found for normal weight and those with no reported chronic disease for participants allocated to Group B compared to those in Group C, such a difference was not significant among those with a nutrition background (Table 2). A significantly higher mean objective assessment score in Group B was also found for participants aged 18–24.9 and 25–59.9, although no specific group differences were found with post hoc analyses. The proportion of individuals that responded correctly to each of the food items displayed can be seen in Appendix A. Between-group significant differences were found in 9 out of the 15 foods displayed, with Group A scoring the highest in two foods (yogurt with fruit, 0% fat, and 100% fruit juice, refrigerated), Group B in five (reduced-salt feta cheese, plant-based alternative to yogurt with fruit, 2% fat plain yogurt, refrigerated fruit drink 0% added sugar, and refined tortilla wraps), and Group C in one (pastry dough, fresh). A significant difference was also found for the refrigerated fruit drink, with a higher percentage of participants allocated in Group A responding correctly, but no significant group difference was found.

Finally, the GLM model of total objective assessment score for each group by self-estimated nutrition knowledge showed no significant differences, meaning that the final score did not differ with nutrition knowledge as reported in any of the groups tested (Figure 5).

## 4. Discussion

The study aimed to evaluate consumers’ food purchasing attitudes and habits and their preferred FOPNL scheme, as well as to explore perceived understanding in relation to objective understanding of nutrition labels based on the latest Hellenic Dietary Guidelines. The results showed that consumers prefer a nutrient-specific and color-coded (i.e., evaluative) scheme. Consumers reporting high nutrition knowledge subjectively stated that they were aware of these schemes, but when objectively assessed, the evaluative summary FOPNL scheme was found most effective for all participants, with a few exceptions for foods high in sugar or fat. The results were consistent by age group and normal weight status and for healthy participants, and the mean objective assessment score did not increase with higher levels of self-estimated knowledge of nutrition guidelines. Overall, the mean objective assessment score was low for all groups assessed, indicating low understanding of both FOPNL schemes and the Nutrition Declaration Table and hence the need for population-specific nutrition education programs.

Consumer behavior surveys and their awareness of FOPNL schemes are essential to understand the consumer attention in this area, since these elements are crucial for successful food choice guidance. A large proportion of consumers reported selecting products based on the food’s name or nutrition claims. This is an important finding, since studies have shown that most consumers spend little time evaluating a product [30], especially when time constraints are applied [31]. The fixation duration has been reported in studies that have used eye tracking devices to examine the first fixation and duration for various FOPNL schemes, with the assumption that duration (or dwelling time) is an index that can relate to the difficulty of the consumers in understanding or the time they require to process the label. It has been reported that warning labels and MTL labels have shorter fixations compared to Guideline Daily Amount (GDA) labels, such as NutrInform Battery, unrelated to the products’ healthiness [30,32]. However, MTL had longer fixations when compared to a healthy choice symbol [31]. Visual attention to health labels, however, did not adequately predict the subsequent choice [31], which raises the question of consumer’s understanding of the FOPNL scheme or the effectiveness of the FOPNL schemes to date, in agreement with this survey’s results, which found a low score in all groups assessed. Overall, the more information provided, the more skills that are required to do the math prior to selecting a healthier food choice, as shown by Prevost et al. through Functional Magnetic Resonance Imaging, testing the neural correlates of evaluating healthfulness [33]. The authors found that healthiness evaluations were faster when consumers were presented MTL information compared to GDA, and more brain regions were active when more than one piece of information was presented, indicating potentially complex processing [33]. However, this brings the questions: to what extent should the label convey interpretive information, and should FOPNL schemes be directive, (i.e., summary labels) or non/semi-directive (nutrient specific). It has been found that the former appeals to consumers who prefer to choose if they want to comply with the information provided, and the latter appeals to consumers that do not mind doing mental work [34]. Their understanding following the mental work is less clear.

The FOPNL scheme reported as preferred by the participants in the present study, was somewhat contradictory compared to the results from the objective assessment, although the objective assessment of understanding focused on only two schemes. Our finding that MTL (which is nutrient specific) was selected as the preferred scheme by most of the respondents is surprising, since MTL is not currently used on products in the Greek market. However, Greek consumers have a higher degree of exposure to nutrient-specific labels, such as RIs or ENL, as observed in other southern European studies [15,23], compared to summary indicator labels. It is important to underline that 7 in 10 respondents reported having observed this on the front of the package, compared to 5 in 10 consumers for the summary indicator labels, such as NutriScore or Healthy Choice. This was consistent among all demographic variables, nutrition background, weight, and disease status, but not among different groups of varying educational levels. More specifically, 40% of individuals with elementary-level education selected Healthy Choice, a summative evaluative scheme. Educational background has been found to play a role, with higher-educational-level consumers showing a preference for clear and color-coded nutritional information [35], but the preference among these individuals was not evaluated. Other than the subjective preference, in terms of the greatest understanding of the 15 foods displayed, the group that was randomly allocated to NutriScore had the best understanding, in total and when aggregated in groups in terms of age, health status, and nutrition knowledge background. These results agree with other studies [4,13,26,36], where NutriScore was most efficient in improving consumers’ food choices based on their nutritional quality compared to others, such as MTL and RIs. Another study evaluating consumer preference via an online questionnaire in the US and Mexico reported that people preferred and had a greater understanding of the warning label as compared to MTL, RIs, and others, with which they were more accustomed, other than the nutrition information tables [7]; however, their objective understanding was not examined.

The consumers subjective understanding of nutrition labels and the influence that nutrition claims have on their purchases may be a concern in terms of the effect of any FOPNL scheme for healthier choices. No differences were found between subjective self-perceived knowledge of nutrition guidelines and total objective assessment score, in any of the groups assessed, indicating that surveys reporting only on consumers’ subjective understanding and preference should be interpreted with caution. Also, based on this study’s results, almost 4 in 10 purchase products based on nutrition claims, and their decision is influenced by the presence or absence of nutrients, irrespective of their awareness of the FOPNL schemes. An online consumer study in Canada evaluated this using various labels and the effect of nutrition claims [8]. Consumers were randomized to different FOPNL groups and were shown four drinks with and without nutrient claims. No differences were found in any of the groups, but only beverages were evaluated. More studies evaluating FOPNL schemes in various regularly consumed foods in relation to the effect of nutrition claims are warranted. To our knowledge, no such studies have been conducted to date that have evaluated these aspects, and selecting “healthy” food can be challenging without a good understanding of the nutritional matrix of a food.

The notion of “healthy” food can be quite perplexing, either due to specific consumer beliefs or interpretations from the nutrition label. Consumers’ beliefs can be a challenge in the field of nutrition, since a healthy food often comes with “ad lib” consumption, such as fruit juice and EVOO in this study. Pertaining to olive oil, 50% of the consumers enrolled in this study responded that it can be consumed ad libitum, with a significantly higher number from the NutrInform Battery group, and the least from the NutriScore group. Fruit juice (100%), however, was correctly categorized as per consumption by the Nutrition Declaration Table group and was misclassified by 42% of the respondents with ad lib consumption from the NutriScore group. In the case of yogurt and whole wheat tortillas, a higher percentage of the participants in the NutriScore group reported ad lib consumption based on the A categorization, although it is recommended that higher-fat dairy and foods high in salt should be less frequently selected and consumed. These are examples of potential healthy bias leading to overconsumption, which is essential to mitigate for an FOPNL scheme to be effective. The “health belief model” for plant-based foods, products usually low in fat and caloric density, is another interesting area this study considered, where an evaluative symbol can help lead to the correct decision. Plant-based alternatives to yogurt scored 7–10% higher with NutriScore compared to the Nutrition Declaration Table and Nutrinform Battery. These examples underline the necessity to increase consumers’ understanding of the FOPNL scheme they will be exposed to, mandatory or not.

Although energy and its close proxy, fat, are well understood [6], salt and sugar may be elements consumers tend to overlook or not comprehend if a nutrition claim is not clearly indicated, despite their associated health risks. This may be due to the low nutrient density of sugar or because salt does not contribute directly to the caloric intake. For high-fat foods with a combination of ingredients, such as reduced-salt feta cheese, the percentage that correctly responded decreased, with the most effective being NutriScore. An FOPNL scheme that does not clearly provide information on these substances or is not easily understood by consumers is concerning, especially in recent years, as consumers may perceive foods containing large amounts of salt as healthy. Specifically, a survey that evaluated plant-based meat products available in the UK found that three quarters of these did not meet the current UK salt targets [37], and another found that 56% of plant-based dairy substitutes from three regions contained more than 5% of the recommended DRV [38], although all products evaluated were lower in energy and SFA compared to their regular counterparts. The high salt content in these foods has also been shown in another study, which found that plant-based meat and dairy substitutes contributed 16% of total salt intake in vegetarians, and this was highest in vegans (17.7%), 90% of which consumed these foods [39].

The proportion of individuals who reported specific chronic diseases is similar to the data reported by those in the Hellenic National Nutrition and Health Survey (HNNHS). This is essential to account for, since nutrition education campaigns should be personalized to the characteristics of the target population. The significant differences between age groups seen in personal, demographic, lifestyle, and medical history data show that FOPNL schemes need to be examined further by age group, since these variables can affect consumer perception. Also, educational campaigns need to be employed to increase consumer awareness of healthier food choices. FOPNL schemes overall need to draw consumers’ attention. A colorful scheme can increase the chances of this [33], and the totality of nutrition information provided should be easily understood by consumers, especially those with low health and nutrition status or overall literacy. Individuals with hypertension may prefer to include in their diet only foods low in salt. Being able to easily select such foods is of great importance, especially with increasing age; in this study population, 1 in 10 adults in total and almost 4 in 10 older adults reported being hypertensive. However, only 22.4% to 30% of consumers identified the correct response to questions about reduced-salt cheese and 22.7% to 28.9% for whole wheat tortillas. although they are of higher salt content. NutriScore was most efficient for reduced-salt feta cheese, whereas NutrInform Battery scored highest for the whole wheat tortillas, an example that again underlines the nutritional matrix interplay with the potential overall healthy bias of FOPNL schemes.

Advocates against an evaluative FOPNL scheme claim that such systems are discriminative, patronizing, and contrary to the principle of empowerment or the concept of personalized nutrition. On the other hand, those in favor of an evaluative scheme believe that the non-interpretive FOPNL is difficult for consumers to understand, while the use of color can capture attention and help to make quick decisions when choosing between foods belonging to the same food category [40] as well as favorably influence portion size judgments for less healthy foods [41]. Furthermore, it has been postulated that FOPNL schemes may also encourage the reformulation of processed foods towards healthier options [12,42], although nutrition claims can lead consumers to unhealthful choices or ad lib consumption. However, although FOPNL’s aim is to improve food purchases, the overall effectiveness remains debatable [43], as shown by the limited consumer understanding in this study. A systematic review that summarizes results from all studies may help clarify grey areas, especially addressing different consumer profiles.

### Study Limitations

Overall, online surveys, although effective and valuable, are affected by generalizability bias. In this specific survey, the results cannot be generalized to older consumers with absolute certainty, since the study failed to achieve an adequate sample among this age group. They can be used to understand the perception and understanding of younger adults, although this must be done with caution, since responders are usually those that are well informed and opinionated. Another limitation is that consumers were not asked to report whether they purchased or consumed the food products they rated, which may have influenced their response rate in any direction. In addition, the level of knowledge of nutrition and dietary guidelines was self-perceived/self-estimated and not assessed with a validated questionnaire. Due to the researchers’ academic background, nutrition students enrolled in predefined data panels also received the email survey link, which may have influenced the findings. However, aggregated data analysis showed the same results for this specific group, other than a slightly higher mean objective assessment score, compared to the total population. Lastly, our study was carried out before the publication of amendments on the algorithm of NutriScore, which resulted in some changes in the classification of some foods (e.g., olive oils from C to B).

## 5. Conclusions

Front-of-pack nutrition labelling (FOPNL) is suggested as a method to promote health by enhancing the consumer knowledge of packaged goods, therefore facilitating healthier decision making [44]. Nevertheless, this study demonstrates the importance of providing nutrition education before introducing label awareness. In addition to implementing a well-designed nutrition education program, it seems that consumers can enhance their understanding of the nutritional value and suggested consumption patterns of food using an evaluative Front-of-Pack Nutrition Labelling (FOPNL) system. On the other hand, persons who have been diagnosed with dyslipidemia, hypertension, or other chronic conditions could benefit from an enhanced FOPNL system if they are provided with sufficient education. This study concluded that both NutriScore and the NutrInform Battery failed to improve objective comprehension when compared to the Nutrition Declaration Table. Comprehending how consumers perceive and behave is essential when analyzing the role and influence of FOPNL schemes in their decision-making processes. Having a subjective concept and knowledge of nutritional guidelines does not ensure a sufficient comprehension of FOPNL. This is because the nutritional composition of food can be complex, which may lead to health views taking precedence over the information provided on the label. Hence, it is crucial for consumers to have a clear and accurate comprehension of FOPNL schemes. While the study determined that an evaluative FOPNL was more efficient, it highlighted an important lesson for nutrition experts: calorie density is not the single factor to consider. It is imperative to provide education regarding the sugar and salt content of food, irrespective of its calorie level.

## Figures and Tables

**Figure 1 nutrients-16-01751-f001:**
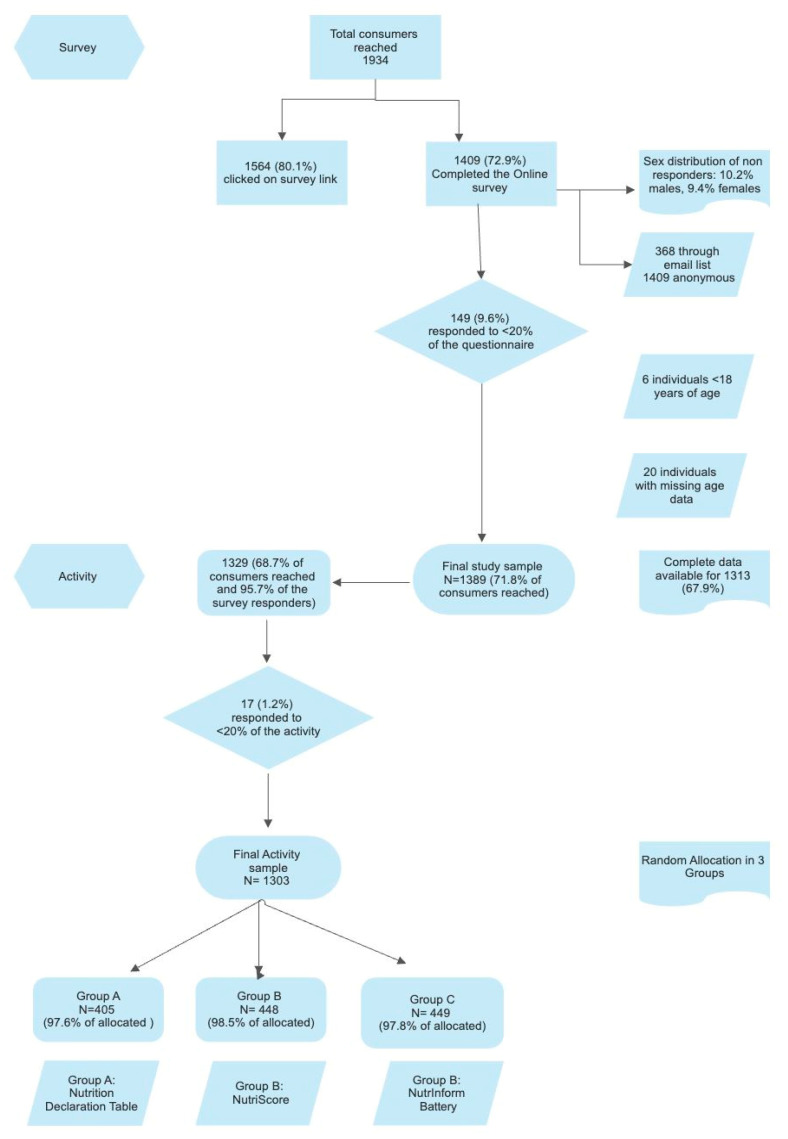
Consumer online survey and objective assessment enrollment flow chart.

**Figure 2 nutrients-16-01751-f002:**
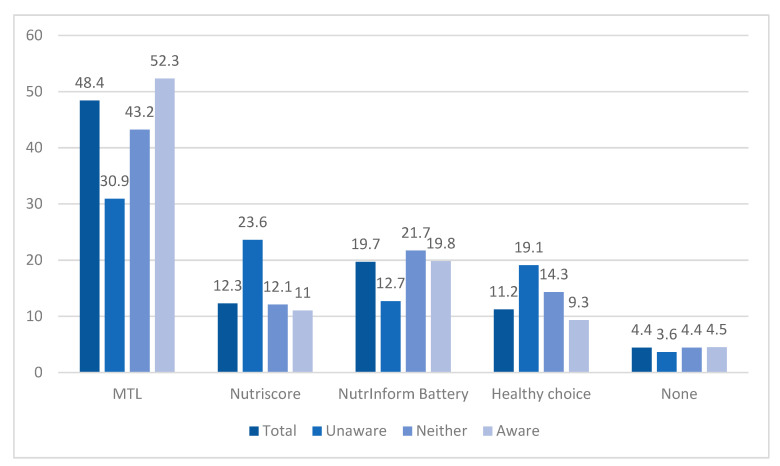
Reported preferred FOPNL scheme presented in total and by level of FOPNL awareness. Unaware: those that reported that they strongly disagree or disagree regarding the statement “are you aware of the FOPNL schemes?”. Neither: those that reported that they neither agree nor disagree regarding the statement “are you aware of the FOPNL schemes?”. Aware: those that reported that they agree or strongly agree regarding the statement “are you aware of the FOPNL schemes?”.

**Figure 3 nutrients-16-01751-f003:**
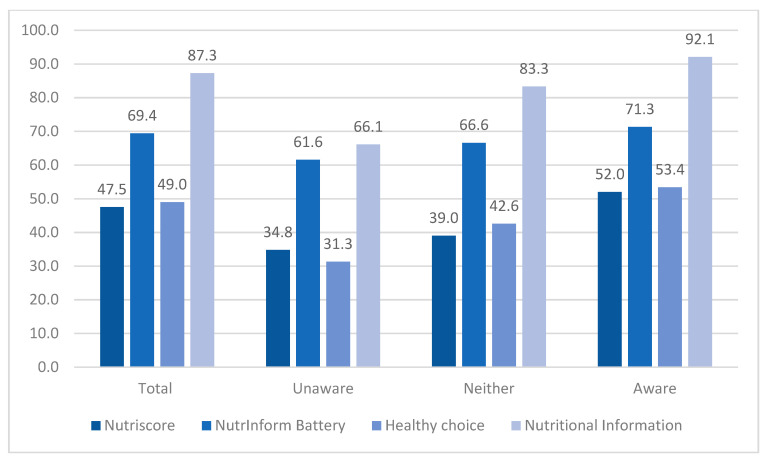
Proportion of individuals that observed the FOPNL schemes on food packages in total and by level of FOPNL awareness. Unaware: those that reported that they strongly disagree or disagree regarding the statement “are you aware of the FOPNL schemes?”. Neither: those that reported that they neither agree nor disagree regarding the statement “are you aware of the FOPNL schemes?”. Aware: those that reported that they agree or strongly agree regarding the statement “are you aware of the FOPNL schemes?”.

**Figure 4 nutrients-16-01751-f004:**
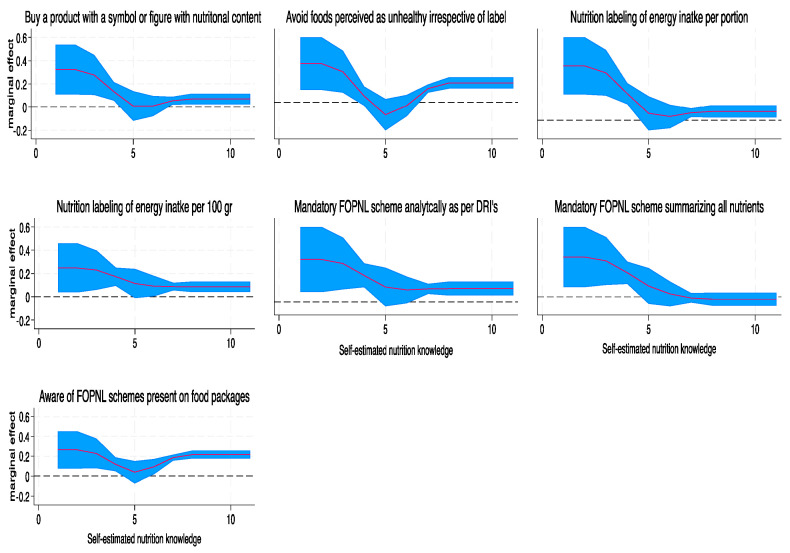
Marginal effects of consumer behavior and FOPNL scheme preference by self-estimated nutrition knowledge. Results from GLM model, adjusted for age group, sex, educational level, occupational status, and nutrition-related degree. Perceived self-estimated nutrition knowledge was based on a 10-point Likert Scale. FOPNL: Front-of-Pack Nutrition Label.

**Figure 5 nutrients-16-01751-f005:**
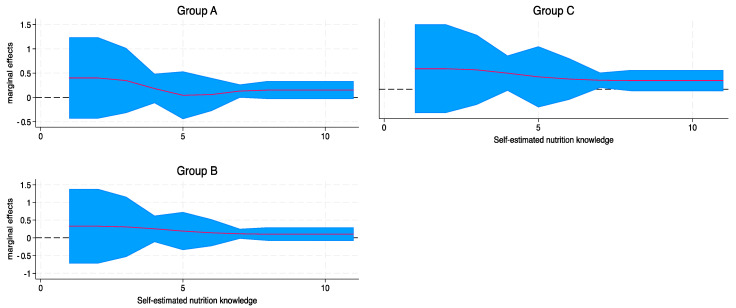
Marginal effects of final objective assessment score by self-estimated nutrition knowledge, per allocated group. Results from GLM model, adjusted for age group, sex, educational level, occupational status, and nutrition-related degree. Perceived self-estimated nutrition knowledge was based on a 10-point Likert scale.

**Table 1 nutrients-16-01751-t001:** Description of FOPNL labels used in the study.

Nutrition Label	Schematic Example	Type	Description
NutriScore	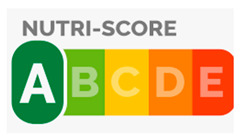	Evaluative—Summative colored	The system uses a five-color scale, ranging from dark green (A) to red (E), to indicate the healthiness of a product. Each letter represents a different score. The score is calculated based on a nutrient profiling system that considers both positive (fiber, protein, and the presence of fruits, vegetables, legumes, and nuts) and negative nutrients (saturated fat, sugars, sodium, and total energy) for 100 g of product.
NutrInform Battery	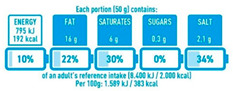	Reductive—Non-Evaluative monochromatic	Monochromatic, “nutrient-specific” scheme per serving size. Numerical information for energy, fat, saturated fat, sugars, and salt from the mandatory nutrition declaration are repeated in a neutral, non-evaluative way.
Multiple Traffic Lights (MTL)	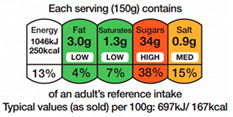	Evaluative—Analytic colored	Color-coded “nutrient specific” per serving, including information on total fat, saturated fat, sugars, and salt/sodium. Each nutrient is displayed separately with its corresponding color code, providing a snapshot of the product’s overall nutritional quality.
Healthy Choice	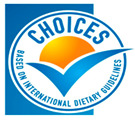	Evaluative—summative	A positive “endorsement” logo/symbol per 100 g of product displayed only on those products complying with certain nutritional criteria. To earn a “healthy choice” label, a product typically includes limits on certain nutrients (total fat, saturated fat, sodium/salt, and added sugars).

**Table 2 nutrients-16-01751-t002:** Main sociodemographic characteristics of sampled population, by age group.

	Total	18–24.9	25–59.9	60+	*p*-Value
Sample, n (%)	1389	311 (22.4)	872 (62.8)	206 (14.8)	
Sex, % female	914 (66.3)	229 (75.3)	569 (65.5)	116 (56.6)	0.292
Area, n (%)					0.003
Attiki & Central Macedonia	1114 (80.3)	230 (73.9)	703 (80.8)	181 (87.9)	
Mainland	195 (14.1)	58 (18.7)	121 (13.9)	16 (7.8)	
Islands and Crete	78 (5.6)	23 (7.4)	46 (5.3)	9 (4.4)	
Educational level					<0.001
Middle school	22 (1.6)	0 (0.0)	4 (0.5)	18 (8.9)	
High school	171 (12.3)	10 (3.2)	113 (13.0)	48 (23.6)	
Higher education	1192 (86.1)	301 (96.8)	754 (86.6)	137 (67.5)	
Degree in nutrition, % yes	399 (28.8)	90 (29.0)	274 (31.5)	35 (17.2)	<0.001
Occupational status, n (%)					<0.001
Unemployed	73 (5.1)	8 (2.6)	59 (6.8)	6 (3.0)	
Employed FT	744 (54.1)	14 (4.6)	673 (77.6)	57 (28.2)	
Employed PT	92 (6.7)	26 (8.5)	63 (7.3)	3 (1.5)	
Pension	154 (11.2)	1 (0.3)	18 (2.1)	135 (66.8)	
University student	313 (22.8)	258 (84.0)	54 (6.2)	1 (0.5)	
Marital status, n (%)					<0.001
Single	649 (46.7)	291 (93.6)	345 (39.6)	13 (6.3)	
Married/cohabiting	598 (43.1)	3 (1.0)	457 (52.4)	138 (67.0)	
Divorced	81 (5.8)	2 (0.6)	52 (6.0)	27 (13.1)	
Widower	32 (2.3)	0	8 (0.9)	24 (11.7)	
BMI, mean (sd)	24.4 (5.0)	22.7 (3.8)	25.5 (4.9)	28.1 (5.2)	<0.001
Weight Status, n (%)					<0.001
Healthy weight	779 (56.4)	245 (78.8)	469 (54.2)	65 (31.9)	
Overweight	284 (27.8)	48 (15.4)	258 (29.8)	78 (38.2)	
Obese	218 (15.8)	18 (5.8)	139 (16.1)	61 (29.9)	
Diagnosed with, % yes:					
Hypertension	147 (10.7)	2 (0.6)	62 (7.1)	83 (43.2)	<0.001
Diabetes	59 (4.3)	3 (0.9)	28 (3.2)	28 (14.9)	<0.001
Hyperlipidemia	250 (18.3)	23 (7.4)	149 (17.2)	78 (41.5)	<0.001
CVD	58 (4.3)	3 (0.9)	24 (2.8)	31 (16.4)	<0.001
Cancer	37 (2.7)	2 (0.6)	20 (2.3)	15 (8.1)	<0.001
Smoking Status, n (%)					<0.001
Smoker	306 (22.0)	50 (16.1)	212 (24.3)	44 (21.4)	
Never smoker	880 (63.3)	252 (81.2)	527 (60.4)	101 (49.0)	
Ex-smoker	203 (14.6)	9 (2.9)	133 (15.2)	61 (29.6)	
On special diet, % yes	341 (24.5)	45 (14.5)	211 (24.2)	85 (41.3)	
Vegetarian diet, % yes	68 (4.9)	14 (4.5)	35 (4.0)	19 (9.2)	0.007

*p*-value derived using one-way ANOVA and chi square test for categorical data, at a = 5%; FT: Full Time; PT: Part Time; special diet defined as any diet low in energy (kcal) and/or salt and/or fat that the participants are following.

**Table 3 nutrients-16-01751-t003:** Total objective assessment score by allocated group; data aggregated by age group, disease status, and background in nutrition.

Activity Score				
	Group A ^1^ N = 405	Group B ^2^ N = 448	Group C ^3^ N = 449	*p*-Value ^4^
Total population	5.6 (2.0)	5.8 (2.3) ^b^	5.4 (1.9) ^b^	0.009
By Age group				
18–24.9	5.3 (2.3)	6.0 (2.2)	5.1 (2.1)	0.017
25–59.9	5.7 (1.9)	5.9 (2.3)	5.5 (1.9)	0.047
60+	5.3 (2.0)	4.9 (2.0)	5.2 (1.7)	0.444
By Weight Status				
Normal weight	5.5 (2.0)	5.8 (2.4) ^b^	5.4 (2.0) ^b^	0.021
Overweight	5.7 (2.1)	5.6 (2.1)	5.5 (1.9)	0.774
Obese	5.6 (2.0)	5.7 (2.2)	5.1 (1.8)	0.071
By Background in Nutrition				
Yes	6.0 (2.0)	6.2 (2.3)	5.6 (2.0)	0.071
No	5.4 (2.0)	5.5 (2.3)	5.3 (1.9)	0.127
By Chronic Disease Status ^5^				
Yes	5.7 (2.0)	5.5 (2.2)	5.2 (1.8)	0.134
No	5.6 (2.1)	5.9 (2.3) ^b^	5.4 (2.0) ^b^	0.030

^1^ Nutrition Declaration Table; ^2^ NutriScore; ^3^ NutrInform Battery; ^4^ group comparisons were derived using ANOVA. The model was adjusted for educational level, occupational status, sex, self-perceived nutrition knowledge, and weight status. Between-group differences were derived using the Tukey–Kramer test following the ANOVA model. ^b^ Denotes differences between Group B and Group C. No other differences were found. ^5^ Chronic disease status defined as individuals reporting being diagnosed with hypertension and/or diabetes mellitus (Type 2) and/or hyperlipidemia and/or cardiovascular disease and/or cancer.

## Data Availability

The original contributions presented in the study are included in the article/Appendix A, further inquiries can be directed to the corresponding author.

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
