# Peer review of "The Relation between Consumer Perception and Objective Understanding of Front-of-Package Nutrition Labels (FOPNLs); Results from an Online Representative Survey"

_nutrients, 2024, doi:10.3390/nu16111751_

Round 1
Reviewer 1 Report
Comments and Suggestions for Authors
very interesting study, it took me a while to read it because it is quite long, but it is upon the editor if he is going to ask to short the manuscript.
my main concern is regarding results of the sample selection. I think that this whole part is in fact methodology. in order to obtain some results you first have to select sample. so the whole paragraph from line 258 to 272 should go under materials and methods. in my opinion results start with table 1. when authors actually calculate something and produce some results based on the sample.
also it is stated that participants were invited to participate. we all know that in this modern times when people have so many other obligations it is not easy to find participants on voluntary basis. authors stated certain bias of results, participants being female, highly educated, familiar with the field of nutrition etc. i think that in this discussion it can be added another component of bias and that is the fact that these participants actively wanted to participate, they somehow wanted their opinion to be heard, so generalization of the results might become difficult. i think that conclusion is well written because study give us the trends and some critical points. if we had really general population i think results would be worse.
Author Response
Thank you for your constructive review.

Reviewer 2 Report
Comments and Suggestions for Authors
In my opinion, this study is of adequate quality and deserves to be published. However, it can be improved on several points, which I will now describe:
- If there is no experimental design, it makes no sense to state in the abstract: "Participants were randomly allocated to these groups".
- The different FOPNL schemes should be shown visually by means of simple figures to help the reader to compare the different models. A clear description of the main characteristics of each would also be advisable.
- Considering that a large part of the survey is demographic data, the sample participating in the study is poorly described. It would be advisable to include a population pyramid and to show the distribution of the sample according to socio-demographic variables using histograms or bar charts.
- Figures 2 and 3 should be better presented as histograms with line charts rather than bar charts.
Author Response
We have taken into consideration the Reviewer’s comment, but have decided to maintain the bar graphs since the variables described are categorical, and percentages by FOPNL schemes are depicted. If lines or histogarms were to be included, readers may wrongly interpret the results.

Reviewer 3 Report
Comments and Suggestions for Authors
The study aims to examine the relationship between consumers' subjective perception and objective evaluation of Front of Pack Nutrition Labels (FOPNL). Review results for this study indicate that overall the manuscript is well-written. However, there are some areas that require revision, and the comments regarding these are as follows:
Abstract:
1. The institution conducting the survey and the title of the survey conducted for this study must be specified.
2. Preference for FOPNL according to demographic characteristics should be presented in the research results. Specifically, it is necessary to indicate which independent variables influence FOPNL preferences. Additionally, statistical relationships between individual independent variables such as total population, age groups, weight status, disease status, nutritional background, and three groups (Nutrition Declaration Table; Nutriscore; NutrInform Battery) should be presented.
Introduction:
1. The necessity and purpose of the study are clearly stated.
2. The citation numbers in the main text should be revised from [1] [2] to [1,2], and from [9] [10] to [9,10].
Methods:
1. Representative sampling is important for population-based studies. Although the approach to ensuring the representativeness of the sample used in this study is described, specific details about the sampling framework and method are required. It must be described in detail.
2. The method of random allocation into three groups (Nutrition Declaration Table; Nutriscore; NutrInform Battery) should be described.
3. Description of the validity and reliability verification process during the development of the survey instrument is required.
4. The number of items in the survey instrument and the results of reliability testing for the survey should be presented.
Results:
1. According to the research results presented in Table 2, statistics on the differences between the three groups (Nutrition Declaration Table; Nutriscore; NutrInform Battery) based on independent variables such as total, age groups, weight status, disease status, and nutritional background are described. However, it should additionally specify which of the three age groups is related to which of the three groups (Nutrition Declaration Table; Nutriscore; NutrInform Battery).
2. It is necessary to set one group among the three as the reference category to ascertain differences between groups.
Discussion:
1. Although limitations of this study have been presented, as commented by the reviewer, it is necessary to verify the statistical significance among age groups and discuss these results accordingly. This applies to other variables beyond age as well.
Conclusions:
1. The conclusion should be clearly articulated, based on the research objectives and the findings obtained from the study.
References:
1. References should be adjusted to fit the journal format.
Author Response
Thank you for your constructive review. Corrected.

Reviewer 4 Report
Comments and Suggestions for Authors
The article presents interesting findings, it is in line with the formats, scope, and depth of the journal, and has the potential to attract its wider audiences. Although food labeling is a topic researched for several decades, namely by marketing scholars, its relevance to managers is increasing, and it is particularly important to find effective ways to communicate with consumers, especially in aspects related to nutrition, which can have important implications for the society as a whole, namely through public health.
As such, I read the article with interest.
The article is also well organized and is easy to read and to follow.
However, the article fails to make a strong impact in the abstract, and there are some missing components in the introduction and conclusion that limit its ability to contribute to the state of the art in its current form. For that reason, I recommend the following improvements.
1. The current version of the abstract is not sufficiently informative and compelling. I suggest that you make it shorter and more concise, and focused not on statistical results, but on overall findings and implications. A more effective abstract would be:
a) shorter,
b) starting by defining the research gap/problem and its current relevance,
c) briefly describing the method, including the sample dimension (the fact that the sample is representative of the population is one of the strengths of the study, and is already mentioned in the abstract),
d) summarize the key findings, but without indicating statistics and avoiding an excessive use of acronyms;
e) clearly highlight the contributions of the article, both theoretical and managerial, and/or highlighting the innovativeness of the article in light of the state of the art.
2. For a manuscript that does not present a literature review as an autonomous section, and instead integrates it in the introduction, the introductory section seems too short. And this is for two reasons:
a) the manuscript needs to provide a clear and complete summary of the state of the art, briefly reviewing the most relevant articles on the topic:
b) an introduction must comprise several components, namely: the current relevance of the topic (this is fairly done at the beginning of the manuscript), the research gap (this clearly needs to be further developed, and the gaps must be clearly stated and substantially supported by extant literature), the introduction also need to explain how the article fills the gap (this is fairly done in the manuscript), and it should highlight the main contributions of the article.
3. The method section needs to be improved.
a) it has organization issues. For instance line 201 is apparently a subtitle but is not numbered or formated as such. It mentions figure 2, but the figure presented after that paragraph is figure 1. The title of section 2.2. needs at least one more word (like procedures or techniques at the end of it).
b) the section is too long, although having missing parts. The subsection 2.1. ould be substantially improved, becoming more concise without loosing information.
c) The missing subsection is "Participants". This component is briefly explained at the start of section 3. But it would be more logic to follow the traditional organization (2. Method, 2.1. Materials, 2.2. Participants, 2.3. Statistical analysis techniques).
d) overall, the method section needs to be concise yet presenting full detail on the procedures, enabling the replication of the study by the readers. Pleas consider this when doing the improvements.
4. The results section is the most important part of this article. I do consider it too long, but I understand the choices made by the authors and will not make any suggestions to this section. I would probably remove (or at least reduce) figure 1.
5. Regarding the conclusion section, this section falls short, given the richness of the findings. My suggestion to the authors is: imagine that a manager, a policy maker, and a researcher decide to read your title, then the abstract, then the conclusion. None of them have much time to read. The conclusion should be improved for these readers. Create four subsections (5.1. Theoretical contributions, 5.2. Managerial implications, 5.3. Societal and policy implications, 5.4. Limitations and future research directions) and develop these sections adequately. The limitations part does not belong to the discussion. And it needs further development.
6. Finally, when making these improvements, make sure that the discussion and conclusions (including all the contributions and implications) are perfectly aligned with your results and supported by them.
I hope it helps! Good luck!
Comments on the Quality of English Language
Only minor issues, overall the text is easy to understand and to follow.
Author Response
Thank you for all your constructive comments. We hope our edits have improved our manuscript.

Round 2
Reviewer 1 Report
Comments and Suggestions for Authors
this is extensive and detailed work worthed publishing. my only concern is full names of supermarkets, and different brands. I did not understand if authors have permission for that or if that is regulated somehow with ethic committie. in my previous work we were asked by journal to call products as product 1, or drink 1, or energy drink 1. if journal does not have issue, i accept everything.
also, talking about overall merit, i put average because , this products are changing very quickly, they get on the market and off the market sometimes in a year, that is why i think authors should discuss a little bit more paradigm of the market, of the false health declarations ...
Author Response
This is extensive and detailed work worthed publishing. my only concern is full names of supermarkets, and different brands. I did not understand if authors have permission for that or if that is regulated somehow with ethic committie. in my previous work we were asked by journal to call products as product 1, or drink 1, or energy drink 1. if journal does not have issue, i accept everything.
Authors: Thank you for acknowledging the work that has been put into this study. We would like to reassure the Reviewer that in under no circmstances were Supermarkets and/or Brand Names inluded in the icons depicted to the participants. All tags (Nutrition Declaration Table and FOPNL scheme) were based on randomly chosen foods found in the market from the food groups included, but no icon was shown of the actual food – only the nutritional information according to Group.
also, talking about overall merit, i put average because , this products are changing very quickly, they get on the market and off the market sometimes in a year, that is why i think authors should discuss a little bit more paradigm of the market, of the false health declarations ...
Authors: we completely understand the Reviewer’s comment, however we required specific inputs from current foods marketed as per the consumers understaning. This is why foods from food groups that are usually consumed were chosen and not those newly marketed. The overall understanding of FOPNL was assessed and not FOPNL per marketed food or brand name.
Furthermore false calims were not discussed since claims are strictly forbidden in EU and were hence not included in study aims. If false claims are included mandatory measures by the Hellenic Food Authority have to be taken.

Reviewer 3 Report
Comments and Suggestions for Authors
The manuscript was appropriately revised based on the reviewers' comments. You did a really good job.
Author Response
Thank you.
Reviewer 4 Report
Comments and Suggestions for Authors
Thank you for your detailed response.
Most of the comments made were addressed, others weren't. Any scientific study needs to adequately support the tested hypotheses. The literature review is too short to support the study adequately and to help discuss the results.
The limitations of the study should not be part of the discussion. The numbering of the title is not correct and actually seems odd to have that subtitle there. The conclusion is not strong enough. The article should provide clear contributions to the literature, implications for managers, policy makers and the society.
The Authors claim that the suggestions made by reviewers did not comply with the formatting rules of the journal. It is up to the editors to suggest the best way to address the pending issues.
Overall, the study is interesting and relevant.
Best wishes!
